# The Molecular Mechanism of Natural Products Activating Wnt/β-Catenin Signaling Pathway for Improving Hair Loss

**DOI:** 10.3390/life12111856

**Published:** 2022-11-11

**Authors:** Dong Wook Shin

**Affiliations:** College of Biomedical and Health Science, Konkuk University, Chungju 27478, Korea; biocosmed@kku.ac.kr; Tel.: +82-430-840-3693

**Keywords:** natural products, Wnt/β-catenin, hair loss, molecular mechanism

## Abstract

Hair loss, or alopecia, is a dermatological disorder that causes psychological stress and poor quality of life. Drug-based therapeutics such as finasteride and minoxidil have been clinically used to treat hair loss, but they have limitations due to their several side effects in patients. To solve this problem, there has been meaningful progress in elucidating the molecular mechanisms of hair growth and finding novel targets to develop therapeutics to treat it. Among various signaling pathways, Wnt/β-catenin plays an essential role in hair follicle development, the hair cycle, and regeneration. Thus, much research has demonstrated that various natural products worldwide promote hair growth by stimulating Wnt/β-catenin signaling. This review discusses the functional role of the Wnt/β-catenin pathway and its related signaling molecules. We also review the molecular mechanism of the natural products or compounds that activate Wnt/β-catenin signaling and provide insights into developing therapeutics or cosmeceuticals that treat hair loss.

## 1. Introduction

Hair loss is a common medical problem that occurs due to both hair density loss and the thinning of hair. This phenomenon is caused by hormonal and non-hormonal problems [1,2]. Hair loss due to the excessive production of dihydrotestosterone, a metabolite of testosterone, is known as androgenetic alopecia (AGA), which has long been extensively studied to understand its pathophysiology [1,2].

Currently, both minoxidil and finasteride, which are approved by the Food and Drug Administration, contribute to promoting hair regrowth or preventing hair loss. In total, 2–3% or 5% topical minoxidil has been used to treat AGA in females and males, respectively [3,4]. As finasteride blocks the conversion process of testosterone to dihydrotestosterone, it has also been utilized to treat AGA in patients. However, these medicines have limited efficacy, requiring continuous administration and causing unpleasant side effects [4,5,6]. Finasteride may cause side effects such as erectile dysfunction in male patients [2,7]. Interestingly, much progress has been made in elucidating the molecular mechanism of hair development and the etiology of hair loss. 

The hair follicle is a complex structure and undergoes a cycle of hair growth (called the anagen phase), regression (called the catagen phase), and rest (called the telogen phase) under the regulation of various signaling pathways: the wingless-type MMTV integration site family member Wnt/β-catenin, transforming growth factor (TGF)-β, fibroblast growth factor (FGF), sonic hedgehog (SHH), bone morphogenetic protein (BMP), and Notch signaling. These various signaling pathways have been implicated in hair regeneration [8,9]. In particular, the Wnt/β-catenin signaling pathway is specifically responsible for hair follicle development, regeneration, and growth [10,11,12].

Based on these molecular targets, novel, active ingredients are required to retard or prevent hair loss. Thus, various studies addressing the increasing problem of hair loss using natural products with few side effects have been examined. Many researchers have identified natural compounds such as polyphenols and natural plants found worldwide as potential active ingredients for treating hair loss [8,13,14,15]. This review discusses the physiological and functional roles of Wnt/β-catenin signaling and its related molecular targets in hair growth. This will also provide meaningful information about various natural compounds and natural plants for treating hair loss, as well as potential insights into developing therapeutics or hair care cosmeceuticals for improving hair loss in the future. 

## 2. Hair Structure and Hair Cycle

Generally, a human scalp maintains hair by regulating the hair cycle [16]. A hair follicle is a specialized organelle that is constantly recycled. Hair growth generally progresses in cycles involving the anagen phase for growth, the catagen phase for cessation, and the telogen phase for rest. The anagen phase generally persists for two to eight years, whereas the catagen phase ranges from two to three weeks. The telogen is finished when the original hair falls out, followed by the generation of new hair [16,17,18]. If the onset of anagen is significantly delayed, it will lead to impaired hair regeneration during the aging process. Hair regeneration is progressed by hair follicle stem cells (HFSCs) [9,19]. Hair follicles extend from the epidermis and reside in the dermis’s deeper layer, forming a bulb-like structure. This bulb surrounds the dermal papillae, which contain dermal papilla cells (DPCs). These cells are an important reservoir of cells for hair regeneration in patients suffering from hair loss [20,21]. Keratinocytes in the hair matrix surround the lower dermal papillae. These keratinocytes proliferate and lead to hair fiber formation during the hair growth cycle. HFSCs are present in the bulge region [22,23]. Hair development is composed of three stages, induction, organogenesis, and cytodifferentiation [23,24]. The induction stage involves epithelial cells thickening to form a placode. The organogenesis stage involves the transfer of signals from the epithelial cells to dermal cells to form a dermal condensate, which then induces the migration of epithelial cells toward the dermis. At the cytodifferentiation stage, the dermal condensate eventually contributes to forming the dermal papillae when it is surrounded by the epithelial cells of hair follicles. 

## 3. Physiological and Functional Roles of Wnt/β-Catenin Signaling for Improving Hair Growth

Hair development involves crosstalk between the epidermal and dermal compartments in the hair follicle [17,25]. The Wnt/β-catenin signaling pathway is a master regulator of hair cells, including outer root sheath cells (ORS), hair matrix cells, and derma papilla cells during hair morphogenesis and the regeneration of hair follicles. Furthermore, Wnt/β-catenin signaling induces the onset of anagen and regulates the hair cycle [10,21,26]. The Wnt/β-catenin signaling pathway also plays an important role in hair morphogenesis and the regeneration of hair follicles [26]. Huelsken et al. demonstrated that genetic ablation of β-catenin failed to cause placode morphogenesis in mouse epidermis [27], implying that Wnt signals are important in hair development. Thus, the Wnt/β-catenin signaling pathway is a potential factor in treating hair loss. 

In the canonical Wnt/β-catenin signaling pathway, the Wnt proteins are associated with the Frizzled receptor and a low-density lipoprotein-related protein (LRP) or receptor tyrosine kinase. This complex inactivates glycogen synthase kinase-3β (GSK-3β) and stabilizes β-catenin to avoid ubiquitin-dependent degradation [26,28]. The stimulated receptor tyrosine kinase inactivated GSK-3β by phosphorylating the upstream kinase extracellular signal-regulated kinase (ERK) or Akt. The stabilized β-catenin translocates to the nucleus and binds to the T cell factor (TCF)/lymphoid enhancer factor (LEF). This complex induces the activation of downstream target genes, which are responsible for cell proliferation and migration [26,28,29].

Dickkopf 1 (DKK1), a Wnt antagonist, inhibits Wnt action by binding to the LRP [30,31]. DKK1 is secreted from DPCs in response to DHT secretion, and it is known to cause and aggravate AGA. In one study, injecting neutralizing DKK-1 antibody retarded the transition of anagen to catagen in mice. In addition, administration with recombinant DKK1 suppressed the Wnt-mediated activation of β-catenin [32]. In another study, an increase in DKK1 was observed in the lesional scalp biopsies of patients suffering from alopecia compared with a healthy control group [33]. R-spondin 1 is a secretory agonist of Wnt/β-catenin, known to antagonize the function of DKK1, including DKK1-mediated hair follicle suppression. The expression and secretion of the Wnt protein are regulated by Wntless (WIs), a transmembrane cargo protein that contributes to the development of hair placodes and hair follicles [34,35]. Another previous study demonstrated that the genetic knockout of epithelial Wls suppressed the formation of hair placode and downregulated the expression levels of BMP2, BMP4, and SHH [36]. Numerous Wnt proteins are responsible for hair growth and regeneration (Table 1). 

### 3.1. Wnt1a

A previous study examined the effects of bone marrow mesenchymal stem cells (BM-MSCs) overexpressing Wnt1a on the regeneration of depilated mouse hair follicles [37]. The treatment of conditioned medium (CM) secreted from overexpressed Wnt1a cells induced hair follicles to undergo from the telogen to the anagen and increased the expression level of alkaline phosphatase (ALP) in the DP area. Interestingly, the Wnt-CM treatment recovered the dihydrotestosterone-induced downregulation of hair induction-related genes, such as Lef1 and Gli1, and enhanced the proliferation level of DPCs [37].

### 3.2. Wnt3a

Kishimoto et al. demonstrated that the overexpression of Wnt3a stimulated β-catenin activity and enhanced hair growth in nude mice with transplanted skin containing DP and keratinocytes [10]. Stearoyl-CoA desaturase 1 (SCD1) is one of the key regulators of lipid and energy metabolism [38]. SCD1 deficiency damages the desaturation of de novo-synthesized palmitoyl into palmitoleoyl-CoA and stearoyl-CoA into oleoyl-CoA. Scd1^−/−^ mice exhibited metabolic waste syndrome and various skin lesions, such as a disrupted epidermal barrier and alopecia. Stoffel et al. found that the absence of palmitoleic acid (9Z-16:1) in the lipid analysis of Scd1^−/−^ mice blocked the posttranslational O-palmitoleoylation of Wnt3a protein and was necessary for stem cell lineage in the developmental progress of the epidermal barrier and hair growth cycle. In the experiment, an artificial lipid barrier was used as a substitute for the disintegrated epidermal lipid barrier, and remarkably, it activated hair bulge progenitor cells and recovered a regular hair growth cycle in Scd1^−/−^ mice [38].

A previous study demonstrated that macrophage-extracellular vesicles (MAC-EVs), including Wnt3a and Wnt7b, enhanced the proliferation rate of DPCs, and elevated the expression levels of Axin2 and Lef1 [39]. Interestingly, MAC-EVs have been found to increase B cell lymphoma 2 (Bcl-2) levels and the phosphorylation level of AKT. Furthermore, the administration of MAC-EV was found to increase hair follicle growth in male BALB/c mice and increase the size of the hair shaft in human hair follicles. These results suggest that MAC-EVs could be clinically used as a potential anagen inducer for treating hair loss [39]. Another report demonstrated that both tumor necrosis factor-alpha (TNF-α) and interferon-γ increased the expression level of major histocompatibility complex class I proteins in DPCs [40]. Furthermore, the loss of the immune response in hair follicles was recovered after treatment with conditioned media (CM) derived from ORS cells. The authors demonstrated that WNT3a-CM, with epidermal growth factor (EGF), can recover hair growth. Autoreactive CD8^+^ T cells were enhanced in alopecia areata patients during the transition from the mid-anagen stage to the late catagen stage. These results suggest that both epithelial and mesenchymal interactions in the hair follicle may regulate the immune response of the hair follicle [40].

### 3.3. Wnt4 

During the anagen phase, the gene expression levels of Wnt ligands—including WNT3, WNT4, and WNT10B—and Wnt target genes—including Axin2 and Lef1—significantly increase in DPCs and secondary hair germ cells [41]. On the other hand, the expression level of secreted Frizzled-related protein 1 (SFRP1), a Wnt inhibitor, diminished. HFSCs upregulate the expression level of WNT4 and WNT10A, meaning these Wnt agonists are essential for activating stem cells [41]. Autocrine Wnt signaling is essential for maintaining stem cell function in murine HFSCs. The downregulation of Wnt ligands or the upregulated expression of Wnt antagonists can cause an uncontrolled murine hair follicle cycle and lead to alopecia [41].

### 3.4. Wnt5a 

Wnt5a belongs to a non-canonical Wnt family and generally antagonizes the function activity of canonical Wnt signaling in other tissues. In hair follicles, Wnt5a is significantly expressed in the bulge region and secondary hair germ cells in the telogen phase [48]. A previous study found that the overexpression of Wnt5a lengthened the telogen stage and diminished anagen entry in mouse dorsal skin [42]. The authors further found that the expression levels of Wnt/β-catenin signaling cascade-related genes significantly reduced after Wnt5a treatment [43]. They also demonstrated that the Wnt5a-induced inhibition of cell proliferation could be recovered using a Wnt3a treatment, implying that Wnt5a suppresses the activation of β-catenin signaling during the regeneration process of hair follicles [43]. A recent report demonstrated that HFSCs exhibit a significant reduction in canonical Wnt signaling and the upregulation of non-canonical Wnt5a signaling during the aging process [44]. The upregulation of Wnt5a in HFSCs enhanced small RhoGTPase Cdc42 activity and caused alterations in the spatial expression of Cdc42 within HFSCs. Aged HFSCs treated with a specific Cdc42 inhibitor called CASIN suppressed the aging-associated increased activity of Cdc42 and restored canonical Wnt signaling. The administration of CASIN in aged mice evoked the onset of the anagen stage and increased anagen skin areas [44]. 

### 3.5. Wnt10b

Wnt10b stimulated the differentiation process of skin epithelial cells into the hair shaft by stimulating β-catenin activity [45]. Interestingly, under serum-free conditions, Wnt10b treatments on whisker hair follicle organ cultures induced the hair shaft elongation via the stabilization of β-catenin [45]. Li et al. demonstrated that the overexpression of Wnt10b in hair follicles induces the biological switch from the telogen phase to the anagen phase, which expresses the structural markers of catagen [46]. In Wnt10b-induced hair follicles, β-catenin was translocated to the nucleus. As expected, the blockade of β-catenin abrogated the functional activity of Wnt10b. These results suggest that Wnt10b contributes to hair follicle growth by shifting telogen to anagen after stabilizing β-catenin [46]. Another study revealed that overexpressed Wnt10b stimulates the proliferation of HFSCs [47]. In addition, cells outside the bulge area began to proliferate; as the Wnt10b-induced hair follicle entered the full anagen phase, canonical Wnt signaling proceeded in hair precortex cells [47]. 

In AGA, androgens impair the differentiation of HFSC by suppressing Wnt signaling [49]. In both spheroids and monolayer culture conditions, DHT reduces the expression levels of Wnt5a and Wnt10b. Conversely, it upregulated the Wnt antagonist, Dkk-1. Interestingly, DPC spheroids reduced the Dkk-1 level and increased the basal expression levels of Wnt agonists. Supplementing DKK-1 into DPC-CM damaged the differentiation of HFSC, similar to the action of androgens. Furthermore, the addition of WNT10b to DPC-CM treated with DHT antagonized the androgen-induced suppression of HFSC differentiation [49].

## 4. Various Factors Interacting with the Wnt/β-Catenin Signaling Pathway

Precise intercellular signaling between epithelial cells and mesenchymal cells is essential for the formation of hair follicles. Several intra- and intercellular signaling molecules crosstalk with the Wnt/β-catenin signaling pathway and play a key role in regulating hair follicles [50,51,52,53,54,55,56,57,58,59]. There are various endogenous factors—such as BMP [50], androgen [51], and growth factors [52,53]—that regulate the Wnt/β-catenin signaling pathway in hair follicles (Figure 1). A previous study reported that dihydrotestosterone diminished Wnt3a-induced TCF/Lef reporter activity in DPCs obtained from AGA [51]. The signal transduction of β-catenin is also modulated by several growth factors signaling pathways. 

BMP signaling suppresses the onset of anagen [50,54]. A previous report demonstrated that the overexpression of BMP6 diminished the proliferation rate of HFSCs and delayed from telogen to anagen in C57 mice [50]. Furthermore, the overexpression of BMP6 suppressed the Wnt10b level in HFSCs. The overexpression of adenoviral Wnt10b down-regulated the number of cells expressing BMP6 compared with the control group. These results suggest that Wnt10b plays a role as an activator, whereas BMP6 plays a role as a β-catenin signaling inhibitor [50]. BMP-deficient stem cells exhibit increased Wnt7a, Wnt7b, and Frizzled 10 receptor expression levels, similar to the molecular profiles of hair germs. In one study, the conditional knockout of BMP-type IA receptor-stimulated proliferation of HFSC [54]. In another study, a treatment using Noggin, a BMP antagonist, led to the activation of EP-SC and the initiation of the anagen phase [54]. 

The administration of epidermal growth factor (EGF) enhanced the β-catenin level in the nucleus and stimulated Wnt10b, β-catenin, and EGF receptor. EGF also elevated the expression levels of survivin, Msx2, and SGK3, which were follicle-regulatory genes. The inhibition of β-catenin activity caused by XAV-939 blocks EGF-induced proliferation in ORS cells [47]. A previous report demonstrated that epidermal growth factor receptor (EGFR) knockout mice failed to develop a hair coat [55]. EGFR plays an important role in attenuating the Wnt/β-catenin signaling pathway during the postnatal hair follicle developmental process, and in the study, EGFR knockout mice exhibited apoptotic cell death in hair follicles, an increase in mitotic activity in matrix cells and a damaged differentiation process in the epithelial lineages for forming hair. EGFR is stimulated in wild-type HFSCs expressing SOX9 or NFATc1, and it is necessary for restraining proliferation and supporting the number of stem cells and their quiescence. In addition, it was found that the expression levels of Wnt4, 6, 7b, 10a, 10b, and 16 increased, and the β-catenin pathway was hyperactivated in the hair follicles of EGFR-knockout mice. These results suggest that a precise balance of both EGFR and Wnt/β-catenin pathways is essential in proliferation and differentiation during the developmental process [55].

Similarly, a previous study revealed that the topical administration of FGF improved hair growth by inducing the initiation of anagen and prolonging the anagen phase in C57BL/6 mice [52]. FGF treatment also induced the activation of β-catenin and Shh in mouse hair follicles. Another study demonstrated that FGF10 enhanced the β-catenin levels in the nucleus and induced the proliferation rates of both ORS cells and DPCs. Conversely, sFRP1 diminished β-catenin levels and suppressed the proliferation rates of both cells [56]. 

Hedgehog (HH) and TGF-β, which modulate epithelial-mesenchymal communication in hair follicles, play roles in the regeneration of hair follicles and hair cycling [57,58,59]. The Wnt signaling pathway is associated with HH signaling pathways [57,58]. Previous studies demonstrated that the Wnt–HH signaling axis is important for maintaining the hair cycle. The activation of β-catenin stimulated the expression level of sonic hedgehog (SHH) expression in the epidermal basal cells of adult mouse skin, indicating that HH signaling is necessary for forming hair follicles [57,58]. Another previous study demonstrated that the TGF-β signaling pathways-related genes were downregulated, whereas the oxidative stress pathway-related genes were upregulated in the bald frontal and haired occipital scalps of patients with androgenetic alopecia [59]. 

The connective tissue growth factor (CCN2) acts as a physiological inhibitor in hair follicle formation and maintains the quiescence status of stem cells. The CCN2 is exclusively expressed in both ORS cells and DPCs. The ablation of CCN2 shortens the telogen phase and enhances the number of hair follicles. On the contrary, the recombinant CCN2 destabilized the β-catenin signaling pathway and diminished the proliferation rate of keratinocytes [53]. 

## 5. Natural Products for Treating Hair Loss by Targeting the Wnt/β-Catenin Signaling Pathway

Many studies have demonstrated that natural plants can stimulate the Wnt/β-catenin pathway or its related biomarkers to treat hair loss (Table 2). 

*Aconitie Ciliare Tuber* (ACT) extract was found to promote ALP activity and the proliferation rate of immortalized DPCs [60]. ACT extract was found to activate the Wnt/β-catenin signaling pathway by elevating β-catenin activity, which stimulated the hair growth of the anagen phase in C57BL/6 mice [60]. *Centipeda minima* (L.) A. Braun and Asch are traditional Chinese medicines [61]. A recent study revealed that *Centipeda minima* (CMX) stimulated the proliferation of human follicle DPCs and increased the expression levels of Wnt5a, Frizzled receptor, and vascular endothelial growth factor (VEGF) [61]. Furthermore, CMX enhanced the phosphorylation of ERK and JNK in DPCs and significantly accumulated β-catenin in a dose-dependent manner. Ginkgo biloba extract, a well-known herbal medicine, is widely used for treating various diseases [62]. Ginkgolide B and bilobalide, the bioactive compounds of c, have been found to promote the growth of hair follicles in American mink. Both compounds elevated the viability of DPCs and induced the secretion level of VEGF. Ginkgolide B upregulated the activities of β-catenin, ERK, and Akt but diminished DKK1 level in DPCs. Bilobalide increased β-catenin and Akt at the molecular level [63]. Another group further demonstrated that water-soluble Ginkgo biloba leaves polysaccharides (WGBP) promote hair growth in alopecia areata mice. They obtained acidic polysaccharides (WGBP-A2) and an RG-I type polysaccharide (WGBP-A2b). WGBP-A2 significantly increased the expression levels of VEGF and hepatocyte growth factor (HGF) but decreased the expression levels of inflammatory factors. WGBP-A2b down-regulated the expressions levels of p-p65, p-IκBα, TNF-α, and IL-1β related to the inflammation signaling pathway in HUVECs [64].

Seed extract from *Malva verticillata* (*M. verticillata*) enhanced Wnt reporter activity and elevated β-catenin levels in human DPCs [65]. Myristoleic acid, a key ingredient of *M. verticillata*, promoted the proliferation rate of DPCs and elevated the transcription levels of HGF, keratinocyte growth factor (KGF), insulin-like growth factor-1 (IGF-1), and VEGF. In addition, myristoleic acid upregulated the phosphorylation levels of both Akt and p38 [65]. This group further isolated effective compounds from *M. verticillata* seed extracts. They identified oleic acid and linoleic acid in the *M. verticillate* (MH)2. Treatment with linoleic acid stimulated Wnt/β-catenin signaling and promoted human follicle DPC growth [66]. Linoleic acid also elevated several growth factors, such as VEGF, IGF-1, HGF, and KGF, in a dose-dependent manner. Linoleic acid suppressed DKK-1. A previous study reported that three plants of the *Polynesian cosmetopoeia*, *Fagraea berteroana*, *Calophyllum inophyllum*, and *Bidens pilosa*, either upregulate *Lef-1* and *PPARD* genes and/or downregulate *DKK1* and *TGFB1* genes [67]. Three natural plants contain various biomolecules, flavonoids, and organic acids, and they significantly elevated the proliferation of DPCs and stimulated hair growth [67]. 

*Prunus mira* Koehne *(P. mira)* seed oil belongs to the *Rosaceae* family [68,69]. *P. mira* nut oil promotes hair follicles into the anagen phase, upregulating the expression levels of Wnt10b and β-catenin. Topical treatments with *P. mira* nut oil significantly enhanced dermal thickness, hair length, and hair weight compared with minoxidil in the depilated dorsal skin of C57BL/6 mice. They demonstrated that *P. mira* could promote hair growth in mice by activating the Wnt/β-catenin pathway [68,69]. The extract of *Polygonum multiflorum* (PM), one of the flowering plants in the buckwheat family, *Polygonaceae*, dose-dependently elevated the cell viability and mitochondrial activity in human DPCs. PM extract significantly increased the expression level of Bcl2 in DPCs and DPC spheroids. PM extract increased VEGF expression but decreased Dkk-1 expression [70]. A red ginseng oil (RGO) treatment significantly restored the regenerative hair capacity in testosterone-treated mice [71], which exhibited a delay of anagen entry. RGO, and its ingredients, linoleic acid, and β-sitosterol accelerated hair growth by inducing an early anagen phase in testosterone-treated mice. The administration of RGO to mice increased β-catenin, Lef-1, cyclin D1, cyclin E, sonic hedgehog, smoothened, and Gli-1 in testosterone-treated mice. RGO also decreased TGF-β level but elevated Bcl-2 level [71].

A previous study found that *Salvia plebeia* R. Brown (Labiatae) has diverse biological activities [72]. *Salvia plebeia* (SP) significantly elevated the proliferation rate of human DPCs compared with the control group. SP increased the expression level of HGF, whereas it decreased the expression levels of TGF-β1 and SMAD2/3. SP stimulated Wnt/β-catenin signaling by increasing the nuclear transfer of β-catenin. SP induced the anagen phase, which resulted in increased hair growth in male C57BL/6 mice [72]. Shallot (*Allium ascalonicum* L.) is a traditional medicine for improving hair growth in Thai folkloric wisdom [73]. Shallot extract downregulated the expression levels of androgen genes such as *SRD5A1* and *SRD5A2.* Conversely, it upregulated the expression level of β-catenin, VEGF, and sonic hedgehogs such as SHH, SMO, and GIL1 and promoted hair growth activity [73]. *Thuja orientalis* (TO) has been used to treat hair loss patients in East Asia. In one study, TO extract stimulated the anagen phase and improved hair growth in C57BL/6 N mice. The TO-extract-treated group increased β-catenin levels and Shh proteins compared with the control group [74]. Watercress is a species of aquatic flowering plant in the cabbage family, *Brassicaceae* [75]. A recent study demonstrated that watercress extract (WCE) could improve hair growth in human hair follicles [75]. WCE significantly decreased DKK1 secretion in the presence of DHT, showing an anti-androgenetic effect. Conversely, WCE increased the production level of R-spondin 1 in DPCs and ORS cells in a dose-dependent manner. WCE-treated hair follicles exhibited 1.6-fold elongation compared to the control. The hair lotion, including 2% WCE, improved hair thickness and density in the six-month clinical trial [75].

Many studies have found many natural compounds that activate the Wnt/β-catenin pathway or its crosstalk, which may be potential therapies to treat hair loss (Table 3). 

In particular, many polyphenol compounds, including flavonoid and chalcone, contributed to countering hair loss by stimulating the Wnt/β-catenin signaling pathway [76,77,78,79,80,81,82,83,84,85,86,87,88,89,90]. Alpinetin is a natural flavonoid compound isolated from *Fabaceae and Zingiberaceae* herbs [76]. The topical administration of alpinetin onto the dorsal skin of depilated C57BL/6J mice was found to stimulate the onset of the anagen stage and retard catagen entry, resulting in an enlengthened anagen phase and longer hair shafts. RNA-seq analysis revealed that alpinetin activated Lgr5+ HFSCs in lower bulges via Wnt signaling. Alpinetin was also found to promote the proliferation of HFSCs, including K15^+^, Lef1^+^, and Gli1^+^ [76]. Baicalin is a glycosyloxyflavone abundantly distributed in various medicinal plants and has multiple biological functions. Xing et al. demonstrated that baicalin significantly induced hair growth in Balb/c-nude mice compared with minoxidil, a positive control. Interestingly, baicalin upregulated the expression levels of Wnt3a, Frizzled-7, β-catenin, and Lef1, whereas it downregulated GSK-3β level compared with minoxidil [77]. Another study demonstrated that baicalin enhanced ALP and Wnt/β-catenin signaling in human DPCs [78]. Additionally, baicalin treatment enhanced the mRNA expression of growth factors, including VEGF and IGF-1. Baicalin rapidly stimulated the telogen phase into the anagen phase compared with the control group. However, baicalin has poor water solubility and limitations on its topical application. Thus, Zeng et al. formulated natural glycyrrhizin (GL) to encapsulate baicalin to overcome its drawbacks [79]. They demonstrated that the optimal GL-baicalin micelle formulations improved the penetration and accumulation of baicalin in the porcine skin without any skin irritation. As expected, this formulation effectively increased the proliferation rate of hDPCs and cellular uptake. Interestingly, this formulation also activated the Wnt/β-catenin pathway and upregulated the expression levels of VEGF and interleukin-10 (IL-10) [79]. 3-Deoxysappanchalcone (3-DSC), a bioactive compound of *Caesalpinia sappan* L. (Leguminosae), is known to possess anti-allergic, anti-inflammatory, and antioxidant properties [80]. A previous study demonstrated that 3-DSC enhances the nuclear translocation of β-catenin and the transcriptional stimulation of the T cell factor [80]. The 3-DSC activated STST3 and elevated FGF and VEGF expression levels. Furthermore, topical treatments using 3-DSC stimulated the anagen phase in C57BL/6 mice [80]. Epigallocatechin-3-gallate (EGCG), a major constituent of polyphenols in green tea, has beneficial effects such as antioxidant properties and anticancer activities [81,82]. EGCG is known to be effective in preventing or treating AGA by selectively suppressing 5α-reductase activity [81]. EGCG increased the proliferation rate of DPCs and promoted hair growth in an ex vivo culture. EGCG significantly increased the phosphorylation levels of ERK and Akt and the Bcl-2/Bax ratio. Similarly, EGCG has exhibited similar effects in the dermal papillae of human scalps in vivo [83]. Kubo et al. screened natural compounds, which stimulated the telomerase reverse transcriptase promoter in HaCaT cells, and they eventually identified fisetin and resveratrol [84]. These polyphenols regulated the expression level of KGF and stimulated the β-catenin pathway. These polyphenols also increased the number of hair follicles and their thickness compared to control mice. [84]. In a subsequent study, the same group further demonstrated that fisetin strongly expressed β-catenin in CD34^+^ cells near hair follicles [85]. Fisetin significantly promoted exosome secretion from HaCaT cells, activating β-catenin in HFSCs and accelerating their proliferation, implying that fisetin could activate the interaction between keratinocytes and HFSCs through exosome secretion, resulting in the promotion of hair growth [85]. Another group reported that resveratrol also enhanced the proliferation rate of hDPCs and prevented H_2_O_2_-induced damage of hDPCs [88]. They also found that resveratrol upregulated the length of the hair shaft and retarded entry into the catagen phase in ex vivo experiments. The topical administration of resveratrol significantly elevated hair growth and accelerated the transition from the telogen phase into the anagen phase in depilated C57BL/6 mice [88]. Quercitrin, a quercetin O-glycoside compound derived from *Houttuynia cordata*, possesses hair growth activity in human DPCs [86]. In one study, quercitrin elevated energy metabolism in human DPCs by increasing mitochondrial membrane potential (ΔΨ) and generating NAD(P)H. The quercitrin treatment increased the Bcl2 level. The quercitrin enhanced the expression levels of growth factors, including FGF, KGF, and VEGF. The quercitrin also increased the phosphorylation levels of Akt, ERK, and CREB in human DPCs, which were reversed by MAPK inhibitors [86]. A previous study revealed that 3,4,5-tri-*O*-caffeoylquinic acid (TCQA), a polyphenolic compound, promoted the hair growth cycle in human DPCs and mouse models [87]. The TCQA treatment completely induced hair regrowth in the shaved area of C3H mice. TCQA upregulated hair growth-associated genes and increased the expression level of β-catenin in vivo and in vitro [87]. Silibinin, derived from *Silybum marianum*, has anti-inflammatory and antioxidant properties that can improve various skin disorders [89]. A previous study demonstrated silibinin-induced 3D spheroid formation in human hair follicle DPCs. The silibinin treatment enhanced Akt phosphorylation level, upregulated Wnt5a level, and stimulated TCF/Lef reporter activity [89]. Troxerutin, a flavonoid derivative of rutin, protects against H_2_O_2_-induced cellular damage in human DPCs [90]. A previous study reported that pretreatment with troxerutin changed miRNAs related to the WNT pathways and the mitogen-activated protein kinase [90].

Several terpenoid compounds have promoted hair growth by stimulating the Wnt/β-catenin signaling pathway [91,92,93]. Loliolide, derived from red or brown algae, elevated the viability of human DPCs without causing cell toxicity [91]. The loliolide enhanced the size of human DPC spheroids and increased the expression levels of IGF, KGF, and VEGF. The loliolide enhanced the nuclear transfer of β-catenin and stimulated TCF/Lef transcriptional activity by activating Akt [91]. Costunolide, which exists in various plants, possesses antioxidant and anti-inflammatory activities [92]. Costunolide upregulated the viability of human hair follicle DPCs and downregulated 5α-reductase activity induced by testosterone. The costunolide treatment also elevated β-catenin levels and cyclin D1 in human DPCs. The administration of costunolide promoted significant hair growth in the depilated dorsal skin of C57BL/6 mice [92]. Oleanolic acid (OA), a pentacyclic triterpenoid compound distributed in various plants, has been reported to promote hair growth [93]. OA treatment elongated the hair shaft and increased an anagen-like stage. The authors showed that β-catenin was highly expressed in the OA-treated groups [93].

In addition, many studies have demonstrated that various natural products effectively activate the Wnt/β-catenin signaling pathway against hair loss [94,95,96,97,98,99,100,101,102,103,104]. Tocotrienol, a vitamin E analog, has antioxidant properties. Ahmed et al. demonstrated that topical treatment with a tocotrienol-rich formulation (TRF) significantly stimulated epidermal hair follicle development and the onset of early anagen in the depilated dorsal skin of mice [94]. In addition, TRF suppressed the expression level of E-cadherin and stimulated the nuclear localization of β-catenin [94]. Valproic acid (VPA) is an anticonvulsant and mood-stabilizing medicine [104]. A previous study revealed that the topical application of valproic acid critically induced hair regrowth and ALP in male C3H mice and increased mean change in total hair count in patients with AGA [95,96]. VPA promoted the Wnt/β-catenin pathway. Another study demonstrated that valproic acid enhanced the viability of human DPCs and ORS cells, stimulated the elongation of the hair shaft, and downregulated catagen transition in an organ culture model [97]. VPA increased β-catenin levels and their nuclear accumulation by inhibiting the activity of GSK-3β in human DPCs [95,97]. Liposomal honokiol, derived from the genus *Magnolia,* is known to have anti-inflammatory and anti-angiogenic activities [98]. A previous study demonstrated that the liposomal honokiol (Lip-honokiol) treatment enhanced hair growth in the shaving area of C57BL/6N mice. Lip-honokiol also stimulated the Wnt3a/β-catenin pathway and downregulated TGF-β1 to induce hair growth in mice [98]. 5-bromo-3,4-dihydroxybenzaldehyde (BDB) is known to possess anti-inflammatory activities in atopic dermatitis model mice and suppresses UVB-induced oxidative stress [99]. BDB elevated the hair fiber length in the vibrissa follicles of rats and the proliferation rate of DPCs. BDB promoted the activation of the Wnt/β-catenin pathway. BDB also significantly upregulated the number of autophagic vacuoles and autophagy (Atg) regulatory proteins such as Atg5, Atg7, Atg16L, and LC3B. Conversely, BDB suppressed TGF-β1-induced Smad2 phosphorylation [100]. Sinapic acid enhanced the secretion level of VEGFs and elevated the proliferation of human DPCs [101]. Sinapic acid also upregulated the phosphorylation level of GSK-3β and the nuclear transfer of β-catenin by activating Akt in DPCs. In one study, myristoleic acid (MA) elevated the proliferation rate of DPCs and activated their G2/M phase by upregulating cyclin A, Cdc2, and cyclin B1 [102]. MA also enhanced the phosphorylation levels of Wnt/β-catenin proteins, such as β-catenin (Ser^552^ and Ser^675^) and GSK3β (Ser^9^). MA enhanced the phosphorylation level of ERK. MA-induced ERK phosphorylation led to alterations in DPC proliferation [102]. Morroniside, derived from *Cornus officinalis*, has various biological properties [103]. Morroniside treatment increased β-catenin levels in the nucleus and elevated the proliferation of ORS cells isolated from the human scalp [103]. In addition, morroniside enhanced the expression level of both Wnt10b and β-catenin in ORS cells. A morroniside treatment elevated the skin thickness and increased the β-catenin level in the epidermis of depilated mouse skin [103].

There were a few clinical studies of these natural products, such as Watercress (Table 2), EGCG [82], and valproic acid [96] (Table 3), for improving hair loss. Unfortunately, these clinical studies did not compare the natural product with two FDA-approved medicines.

## 6. Materials and Methods

### 6.1. Search Strategy

Until 30 October 2022, PubMed searches were conducted for articles that studied the effects of natural compounds on hair growth. To reflect the latest studies, the time frame of the search was limited from the year 2000 to the present (within 22 years). The search combined the keywords “hair growth”, “Wnt“, “alopecia”, “plant”, “polyphenol”, “flavonoid”, “natural compound”, “dermal papilla cells”, and “hair follicle”. 

### 6.2. Selection of Studies

Inclusion criteria are as follows: (1) published in English; (2) intervention included natural compounds or plants; (3) hair growth or alopecia; (4) hair follicle or dermal papilla cells.

### 6.3. Data Extraction

Data about natural products were chosen from selected references (Table 2 and Table 3) as follows: (1) natural plant source; (2) cell or animal type; (3) working concentration; (4) molecular targets (or major molecular mechanism); (5) references.

## 7. Conclusions

Hair loss, or alopecia, is a common disorder observed among people worldwide. The side effects of finasteride and minoxidil have led to many studies elucidating the molecular mechanisms of hair development, the hair cycle, and growth. Among multiple molecular mechanisms, the Wnt/β-catenin signaling pathway is well-known as an essential pathway in hair regeneration. Accordingly, many studies have identified natural products or compounds worldwide as potential active Wnt/β-catenin candidates for treating hair loss. If these natural products reduce the side effects of preventing hair loss, they will be more effective than minoxidil and finasteride. However, since hair regeneration is performed through a precise network between the epidermal and dermal compartments, the combination of Wnt/β-catenin activators and other candidate drugs, such as BMP inhibitors and TGF-β inhibitors, may help treat hair loss. 

Thus, future studies must be clinically applied without affecting efficacy or causing the side effects of these natural products. These studies will contribute to developing therapeutics or cosmeceuticals for improving hair growth. 

## Figures and Tables

**Figure 1 life-12-01856-f001:**
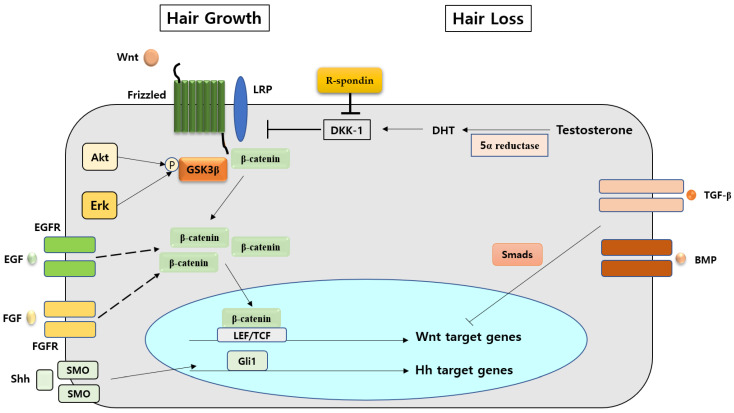
Various factors interact with the Wnt/β-catenin signaling pathway in hair growth and hair loss, respectively.

**Table 1 life-12-01856-t001:** The physiological and functional roles of each Wnt isoform on hair growth.

Wnt Isoforms	Cell or Animal Type	Experimental Methods	Functional Roles	References
Wnt1a	C57BL/6 mice	Overexpression of Wnt1a-CM derived from BM-MSCs	↑ Versican, Lef-1 ↑ Gli-1, Ptc-1 ↑ ALP ↑ Transition of telogen to anagen	[37]
Wnt3a	Nude mouse	Grafting of Chick embryo fibroblasts producing Wnt3a	↑ Hair inductive activity	[10]
	SCD1-null mice	-	↓ Posttranslational O-palmitoleoylation of Wnt3a	[38]
	Balb/c mice DPCs	MAC-EV containing Wnt3a/Wnt7b	↑ Hair Growth ↑ HF Number, Dermis thickness ↑ HF shaft elongation ↑ KGF, VEGF, Axin2, Lef1 ↑ Versican, ALP, p-AKT, Bcl-2	[39]
	Human hair organ DPCs	WNT3a-CM	↑ Hair growth	[40]
Wnt4	Male occipital scalp HFs from patients		↑ WNT3, WNT4, WNT10B ↑ AXIN2, LEF1 ↓ SFRP1, DKK1	[41]
Wnt5	C57BL/6J mice	Overexpression of adenovirus, Wnt5a	↑ Telogen stage ↓ β-catenin ↓ Myc, Axin2	[42,43]
	C57BL/6 mice	Overexpression of adenovirus Wnt5a, and AdSimBC	↓ β-catenin, Lef1	[43]
	C57BL/6 mice Cdc42GAP^−/−^ mice	overexpression of adenovirus Wnt5a,	↑ small RhoGTPase Cdc42	[44]
Wnt10b	whisker hair follicle organ culture from C3H/HeN mice	Medium containing Wnt-10b	↑ Shaft growth ↑ Keratin 15 in the bulge region	[45]
	C57BL6/N mice	In vivo injection of AdWnt10b	↑ Wnt/β-catenin ↑ Hair follicle regeneration	[46]
	C57BL/6 mice	Intradermal injection of AdWnt10b	↑ β-catenin ↑ CD34	[47]

Bone marrow mesenchymal stem cells (BM-MSCs), Conditioned medium (CM), Keratinocytes growth factor (KGF), Vascular endothelial growth factor (VEGF), Stearoyl-CoA desaturase 1 (SCD1), SFRP1 (secreted frizzled-related protein 1), Hair follicle (HF), AdSimBC (siRNA targeting β-catenin), “↑” increased; “↓” decreased.

**Table 2 life-12-01856-t002:** Each natural plant for improving hair loss.

Chemical Name (Active Material)	Cell or Animal Type	Working Concentration	Molecular Targets or Effects	References
*Aconitie ciliare tuber*	Human immortalized DPCs RvDP C57BL/6 mice	5–50 μg/mL 10 mg/mL	↑ Wnt/β-catenin ↑ ALP	[60]
*Centipeda minima* (L.) (CMX) A. Braun and Asch	Human DPCs	0–12.5 μg/mL	↑ Wnt5a, Frizzled, VEGF ↑ ERK, JNK	[61]
*Ginkgo biloba extract* (Ginkgolide B and bilobalide)	Human DPCs	0–100 μM	↑ Akt, ERK ↑ β-catenin ↓ DKK1	[62,63,64]
*Malva verticillata* (Linoleic acid and oleic acid in n-hexane)	Human DPCs	20 μg/mL	↑ Wnt/β-catenin ↑ IGF-1, KGF, ↑ VEGF, HGF ↓ DKK1	[65,66]
*Polynesian cosmetopoeia,* *Bidens pilosa*, *Calophyllum inophyllum*, and *Fagraea berteroana*	Human DPCs	0–12.5 μg/mL	↑ Wnt/β-catenin ↓ DKK1, TGF β	[67]
*Polygonum multiflorum extract*	C57BL6/N mice	4.7 mg/12 cm^2^	↑ Shh ↑ β-catenin	[68,69]
*Prunus mira Koehne*	C57BL/6 mice	Crude oil 15.06–60.26 mg/cm^2^/day	↑ Wnt 10b/β-catenin	[70]
Red ginseng oil (RGO)	C57BL/6 mice	10%	↑ Wnt/β-catenin ↑ Lef-1, Shh/Gli1 ↑ Cyclin D1, E ↓ TGF-β	[71]
*Salvia plebeia* (SP) R. Brown (Labiatae)	Human DPCs C57BL/6 mice	0–31.3 μg/mL 1000 μg/mL	↑ Wnt/β-catenin ↑ Akt, ERK ↑ HGF ↓ TGFβ, SMAD2/3	[72]
*Shallot* (*Allium ascalonicum* L.)	Human DPCs	0.1 mg/mL	↑ Wnt/β-catenin ↑ VEGF ↑ Shh, SMO, Gil1	[73]
*Thuja orientalis*	C57BL6/N mice	5.05 mg/12 cm^2^/day	↑ Wnt/β-catenin ↑ Shh	[74]
Watercress	Ex vivo organ culture of human hair follicles 44 male subjects with AGA	2% WCE-containing Lotion (twice daily for 6 months)	↑ R-spondin 1 ↓ DKK1 ↑ hair thickness (diameter) ↑ hair density	[75]

Dermal papillar cells (DPCs), Sonic hedgehog (Shh), Rat vibrissa dermal papilla cells (RvDP), Red ginseng oil (RGO), Keratinocyte growth factor (KGF), Hepatocyte growth factor (HGF), Vascular endothelial growth factor (VEGF), Insulin growth factor (IGF-1), “↑” increased; “↓” decreased.

**Table 3 life-12-01856-t003:** Each natural compound for improving hair loss.

Category	Chemical Name (Active Material)	Cell or Animal Type	Working Concentration	Molecular Targets	References
Flavonoid	Alpinetin	C57BL/6 J mice	3 mg/mL	↑ Lgr5^+^ HFSCs ↑ Wnt/β-catenin ↑ K15^+^, Lef1^+^, Gli1^+^ ↓ caspase-3	[76]
Flavone glycoside	Baicalin	Balb/c-nu mice transplanted with skin cells of C57BL/6 mice Human DPCs C57BL/6 mice	50 μM or 100 μM GL-baicalin micelle (each 5 mg/ 1 mg)	↑ Wnt/β-catenin ↑ Wnt3a ↑ Frizzled 7, ↑ Lef1, ALP, VEGF	[77,78,79]
Chalcone	3-Deoxysappan chalcone (3-DSC)	Human DPCs C57BL/6 mice	0.1–10 μM 3 mM	↑ Wnt/β-catenin ↑ VEGF, FGF	[80]
Flavanol	Epigallocatechin-3-gallate (EGCG)	Human DPCs ORS cells 44 patients with AGA	0.25–4 μM Lotion containing EGCG for 24 weeks (twice daily).	↑ ERK, Akt ↑ Bcl-2/Bax ratio ↑ median anagen-to-telogen ratio	[81,82,83]
Flavonol	Fisetin	C57BL/6 mice	0.1%	↑ Wnt/β-catenin ↑ KGF	[84,85]
Flavone (Quercetin O-glycoside)	Quercitrin	Human DPCs Cultured hair follicles	0.1–100 nM, 1–10 μM	↑ NAD(P)H, ΔΨ ↑ Bcl-2 ↑ Akt, ERK, CREB ↑ FGF, KGF, VEGF	[86]
Tannin	3,4,5-tri-*O*-caffeoyl quinic acid (TCQA)	Human DPCs C3H mice	10 μM 1%	↑ Wnt/β-catenin ↑ ALP	[87]
Stilbenoid	Resveratrol	C57BL/6 mice	0.1%	↑ Wnt/β-catenin KGF	[84,88]
Flavonoid	Silibinin	3D spheroid derived from Human DPCs	10 μM	↑ Wnt5 ↑ ALP, Akt ↑ FGF7	[89]
Flavonoid derivative of rutin	Troxerutin	Human DPCs	10 μM	↑ Wnt/β-catenin	[90]
Monoterpenoid hydroxyl lactone	Loliolide	Human DPCs spheroid	20 μg/mL	↑ β-catenin ↑ VEGF, IGF, KGF ↑ Akt, ALP	[91]
Sesquiterpene lactone	Costunolide	Human DPCs C57BL/6 mice	0.1–3 μM	↑ Wnt/β-catenin ↑ Gli1 ↓ 5α-reductase, TGF- β	[92]
Pentacyclic triterpenoid	Oleanolic acid	Human hair follicle organ culture	1 or 10 μg/mL	↑ Wnt/β-catenin	[93]
Vitamine E analog	Tocotrienol	C57BL/6 mice	5 mg/cm^2^	↑ β-catenin ↓ E-cadherin	[94]
Organic weak acid	Valproic acid	Human DPCs, ORS C57BL/6, C3H mice 40 patients with AGA	0.1 mM–700 mM A tonic spray containing 8.3% sodium valproate for 24 weeks	↑ Wnt/β-catenin ↑ ALP ↑ The mean change in total hair count	[95,96,97]
Lignan	Honokiol	C57BL/6N mice	20 mg/mL	↑ Wnt3a/β-catenin ↓ TGF-β	[98]
bromophenol	5-bromo-3,4-dihydroxybenzaldehyde (BDB)	Human DPCS	0.01, 0.1, 1 μM	↑ Wnt/β-catenin ↑ Atg5, Atg7, Atg16L, ↑ LC3II ↓ TGF-β	[99,100]
Cinnamic acid derivative	Sinapic acid	Human DPCs	10, 50, 100 μM	↑ Wnt/β-catenin ↑ VEGF, IFG-1 ↑ Akt, ERK	[101]
Omega-5 fatty acid	Myristoleic acid	Human DPCs	1, 5 μM	↑ Wnt/β-catenin ↑ Cyclin A, Cdc2, ↑ Cyclin B1 ↑ ERK, Akt	[102]
Iridoid glycoside	Morroniside	ORS cells C57BL/6 mice	1 or 10 μM 100 μM	↑ Wnt10b/β-catenin ↑ Lef1	[103]

Glycyrrhizin-baicalin micelle (GL-baicalin micelle), 5-bromo-3,4-dihydroxybenzaldehyde (BDB), “↑” increased; “↓” decreased.

## Data Availability

Not applicable.

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
