# Peer review of "The Molecular Mechanism of Natural Products Activating Wnt/β-Catenin Signaling Pathway for Improving Hair Loss"

_life, 2022, doi:10.3390/life12111856_

Round 1

Reviewer 1 Report

The author's paper is very well organized on Wnt/b-catenin signaling in the hair field. Based on many references, the effect of Wnt/b-catenin signaling on hair was well summarized. However, there are some inaccuracies that require correction.

page 2 line 75~77

In the case of alopecia, it is not a symptom of a lack of stem cells. . Also, the hair shaft does not start from the epithelial matrix. Please remove these sentences. And hair development stages are divided into undifferentiated epithelium, placode, germ, peg, and bulbous peg. What the author mentioned seems to be the hair formation stage during regeneration. Therefore, please delete the hair stem cell part. It is unfortunate that the author referred to too many review papers

The author has mentioned many kinds of Wnt family. I would like to organize many parts into a table. At this time, I would like to make a table by dividing human cells and mouse cells.

Author Response

Dear Reviewer 1

Thank you for your feedback.

The author's paper is very well organized on Wnt/b-catenin signaling in the hair field. Based on many references, the effect of Wnt/b-catenin signaling on hair was well summarized. However, there are some inaccuracies that require correction.

page 2 line 75~77

In the case of alopecia, it is not a symptom of a lack of stem cells. Also, the hair shaft does not start from the epithelial matrix. Please remove these sentences. And hair development stages are divided into undifferentiated epithelium, placode, germ, peg, and bulbous peg. What the author mentioned seems to be the hair formation stage during regeneration. Therefore, please delete the hair stem cell part.

Answer) Thank you for your feedback. I made a mistake and follow your direction. Thus, I deleted the problematic sentences, which were erased in red color.

It is unfortunate that the author referred to too many review papers

Answer) Thank you for your feedback. Thus, I changed some reviews with original articles, which were written in a red color

The author has mentioned many kinds of Wnt family. I would like to organize many parts into a table. At this time, I would like to make a table by dividing human cells and mouse cells.

Answer) Thank you for your feedback. Thus, I organize many parts into a Table 1 containing species or cell type, experimental methods, and functional roles.

Reviewer 2 Report

The authors aimed to review the functional role of the Wnt/β-catenin pathway and the molecular mechanism of natural products or natural compounds that activate Wnt/β-catenin signaling in hair follicle. However, here are some problems that must be solved before publication. If the following problems are well-addressed, this reviewer believes that this paper can make a certain contribution to the therapy of hair loss.

1、Your manuscript needs to be carefully edited, especially for grammar, spelling, and sentence structure. Some sentences contain grammatical mistakes, such as, in line 86, "compartments" would be consistent.

2、The basic experiments of natural products that promoted hair growth would be described more concise. It is worth considering showing more summative descriptions in the form of a table.

3、The structure of the article needs to be modified, and there is a lack of points of focus between the Wnt/β-catenin and natural products, which makes these two parts seem to be separated.

4、Whether there are clinical studies that demonstrated the clinical effect of these natural products for hair loss and contrast experiments that compared these natural products with Minoxidil and Finasteride? The relevant clinical significances need to be described.

Author Response

Reviewer 2

The authors aimed to review the functional role of the Wnt/β-catenin pathway and the molecular mechanism of natural products or natural compounds that activate Wnt/β-catenin signaling in hair follicle. However, here are some problems that must be solved before publication. If the following problems are well-addressed, this reviewer believes that this paper can make a certain contribution to the therapy of hair loss.

1、Your manuscript needs to be carefully edited, especially for grammar, spelling, and sentence structure. Some sentences contain grammatical mistakes, such as, in line 86, "compartments" would be consistent.

>> Thank you for your feedback. I fixed some mistakes and decided to get an editing service for a better revision.

2、The basic experiments of natural products that promoted hair growth would be described more concise. It is worth considering showing more summative descriptions in the form of a table.

>> Thank you for your feedback. Thus, I described basic experiments of natural products more concise, and deleted many sentences.

3、The structure of the article needs to be modified, and there is a lack of points of focus between the Wnt/β-catenin and natural products, which makes these two parts seem to be separated.

>> Thank you for your feedback. I agreed with you, and I seriously considered the structure of my manuscript. I checked several kinds of reviews between the target mechanism and natural products. And another reviewer did not point out this problem. To meet two reviewers’ feedback, I emphasized the relationship between Wnt/beta and natural products in Part 5.  

4、Whether there are clinical studies that demonstrated the clinical effect of these natural products for hair loss and contrast experiments that compared these natural products with Minoxidil and Finasteride? The relevant clinical significances need to be described.

>>> Thank you for your feedback. Thus, I searched for PubMed with “finasteride or minoxidil and hair loss and clinical study and natural product or polyphenol, which were written in this manuscript”. Strangely, I found only three papers, but these papers are not involved with Wnt/beta catenin. Thus, I added the following sentences at the end of the result part. The following sentence is “There were a few clinical studies of these natural products such as Watercress, EGCG [102], and valproic acid [103] for improving hair loss (Table 3). Unfortunately, these clinical studies did not compare the natural product with two FDA-approved medicines.”
